# Machine-Learning-Based Classification Model to Address Diagnostic Challenges in Transbronchial Lung Biopsy

**DOI:** 10.3390/cancers16040731

**Published:** 2024-02-09

**Authors:** Hisao Sano, Ethan N. Okoshi, Yuri Tachibana, Tomonori Tanaka, Kris Lami, Wataru Uegami, Yoshio Ohta, Luka Brcic, Andrey Bychkov, Junya Fukuoka

**Affiliations:** 1Department of Pathology Informatics, Nagasaki University Graduate School of Biomedical Sciences, Nagasaki 852-8588, Nagasaki, Japan; sanohisao5584@gmail.com (H.S.); ethanokoshi@gmail.com (E.N.O.); tachibana.yuri@kameda.jp (Y.T.); krislami839@gmail.com (K.L.); 2Department of Diagnostic Pathology, Izumi City General Hospital, Izumi 594-0073, Osaka, Japan; tanaka.t.tomonori@gmail.com (T.T.); yoshio.oota@tokushukai.jp (Y.O.); 3Department of Pathology, Kameda Medical Center, Kamogawa 296-8602, Chiba, Japan; uegami.wataru@kameda.jp (W.U.); bychkov.andrey@kameda.jp (A.B.); 4Department of Pathology, Kobe University Graduate School of Medicine, Kobe 650-0017, Hyogo, Japan; 5Diagnostic and Research Institute of Pathology, Medical University of Graz, 8010 Graz, Austria; luka.brcic@medunigraz.at

**Keywords:** transbronchial lung biopsy (TBLB), non-diagnostic samples, Delphi method, interface bronchitis/bronchiolitis (IB/B), decision-tree based classifiers

## Abstract

**Simple Summary:**

The distinction between entirely benign and potentially mis-sampled cases presents a notable challenge in the histological examination of transbronchial lung biopsy (TBLB) specimens derived from pulmonary nodules that lack tumor or atypical cells. Such cases are often categorized as non-diagnostic. This study aims to develop a machine learning-based classifier for TBLB specimens, with a specific focus on analyzing the micro-environmental histological reactions present in TBLB and correlating these changes to either benign or malignant status, and to avoid unnecessary sampling procedures and lead to prompt treatment initiation.

**Abstract:**

Background: When obtaining specimens from pulmonary nodules in TBLB, distinguishing between benign samples and mis-sampling from a tumor presents a challenge. Our objective is to develop a machine-learning-based classifier for TBLB specimens. Methods: Three pathologists assessed six pathological findings, including interface bronchitis/bronchiolitis (IB/B), plasma cell infiltration (PLC), eosinophil infiltration (Eo), lymphoid aggregation (Ly), fibroelastosis (FE), and organizing pneumonia (OP), as potential histologic markers to distinguish between benign and malignant conditions. A total of 251 TBLB cases with defined benign and malignant outcomes based on clinical follow-up were collected and a gradient-boosted decision-tree-based machine learning model (XGBoost) was trained and tested on randomly split training and test sets. Results: Five pathological changes showed independent, mild-to-moderate associations (AUC ranging from 0.58 to 0.75) with benign conditions, with IB/B being the strongest predictor. On the other hand, FE emerged to be the sole indicator of malignant conditions with a mild association (AUC = 0.66). Our model was trained on 200 cases and tested on 51 cases, achieving an AUC of 0.78 for the binary classification of benign vs. malignant on the test set. Conclusion: The machine-learning model developed has the potential to distinguish between benign and malignant conditions in TBLB samples excluding the presence or absence of tumor cells, thereby improving diagnostic accuracy and reducing the burden of repeated sampling procedures for patients.

## 1. Introduction

Over the last decade, the landscape of early diagnosis and treatment of lung cancer has significantly evolved, mainly due to the increasing adoption of lung cancer screening [1,2,3] and the implementation of video-assisted thoracic surgery (VATS) for managing early-stage diseases [4,5]. Nevertheless, lung cancer remains the leading cause of cancer-related deaths worldwide [6]. Transbronchial lung biopsy (TBLB) is widely used technique to provide a definitive pathological assessment of pulmonary nodules, which are a common manifestation of lung cancer seen on radiology [7].

In the context of histopathological examination, determining the malignant nature of a sample becomes straightforward when malignant cells are present. However, challenges arise when confronted with cases without tumor cells—where only inflammatory cells or fibrosis are detected, especially when a pulmonary nodule is identified on imaging. In such instances, there is limited evidence to support the conclusion of whether a nodule is benign or if it is a sampling error that failed to reach the nodule. Ultimately, such cases are frequently categorized as non-diagnostic samples. Despite a diagnostic accuracy of up to 94% achieved with computed tomography (CT)-guided TBLB, a notable false-negative rate remains persistent [8,9]. In certain scenarios, the target site may be inaccessible, rendering the sampling of the lesion infeasible. Additional diagnostic measures, such as rapid on-site evaluation (ROSE) or bronchoalveolar lavage (BAL), may be employed as supplementary tests [10]. However, even with the integration of these adjunctive measures, a definitive diagnosis is often not achieved.

Lung cancers, along with many other malignant diseases, are histologically characterized not only by the presence of tumor cells but also by various alterations in the surrounding tissue, collectively referred to as a desmoplastic reaction. This reaction involves the expansive remodeling of the tumor stroma around the malignant mass, consisting of fibroblasts, mesenchymal cells, immune cells, blood vessels, and the extracellular matrix [11]. Additional features such as angiogenesis and inflammatory cell infiltration are also associated with changes occurring near a malignant tumor [12]. In similar fashion, inflammatory changes directed towards the airway epithelium are associated with non-malignant airway-related diseases, such as infection [13]. There have been reports of abscess formation, granulomatous inflammation, and chronic inflammation with fibroplasia as negative predictive factors in CT-guided percutaneous biopsy [9,14,15,16]. Recognizing these distinct pathological features in non-diagnostic TBLB specimens can not only improve the diagnostic accuracy but also prevent patients from the need for recurrent invasive investigative procedures.

In recent years, there has been notable progress in the medical field with the increasing adoption of deep learning and machine learning algorithms for computer-aided detection. The early detection of diseases plays a pivotal role in mitigating mortality rates linked to cancers and tumors. To address these issues, in addition to the conventional approaches of machine learning and deep learning, several custom models have been put forth to augment the capabilities in medical diagnostics [17,18,19].

In light of the above, the objective of this study is to scrutinize the histological features discernible in TBLB specimens, aiming to delineate the features indicative of benign or malignant status. Subsequently, the study aims to construct a machine-learning algorithm adept at classifying TBLB samples as benign or malignant, thus contributing to the refinement of diagnostic precision in such challenging scenarios.

In this manuscript, we initially extract cases of TBLB for pulmonary nodular lesions. We then establish histological candidates to be scored using the Delphi method. Following this, we obtain scores from multiple pathologists, develop a machine learning algorithm using these scores, and evaluate the algorithm’s effectiveness in classifying TBLB samples as either benign or malignant.

## 2. Materials and Methods

### 2.1. Patient Cohort

A total of 277 consecutive TBLB cases were collected from April 2022 to February 2023 at the Izumi City General Hospital (Osaka, Japan). Baseline information, such as gender, age, smoking history, biopsy location, and number of biopsy sections, was gathered from electronic medical records (Table 1). Tissue sections were stained with hematoxylin and eosin and were scanned at 20× magnification using a digital slide scanner (Nano Zoomer S210, Hamamatsu Photonics, Hamamatsu, Japan), and whole slide images (WSI) were created and uploaded to the cloud-based system (Mixture Report, N Lab Co. Ltd., Nagasaki, Japan) for review. In the study, informed consent was obtained through an opt-out method. The Centralized Institutional Review Board (IRB) has approved our research protocol, which allows participants to opt in or out (M2021-315).

### 2.2. Pathological Findings and Scoring

In our study, the Delphi method was used to select from the candidates of indicators suggestive of a benign or malignant case. Three pathologists (TT, WU, JF) scored the frequency and expected contribution to the diagnosis on a scale of 1 (very low) to 5 (very high) and chose the six pathological findings with the highest sum: interface bronchitis/bronchiolitis (IB/B), plasma cell infiltration (PLC), eosinophil infiltration (Eo), lymphoid aggregation (Ly), fibroelastosis (FE), and organizing pneumoniae (OP) (Figure 1). Two pathologists (YT and HS) reviewed WSIs through the cloud system and scored the above six findings blinded to the pathological diagnoses and clinical data. IB/B, PLC, Eo, and Ly were scored within the bronchi or bronchioles. FE and OP were scored within the lung parenchyma. If no bronchiolar epithelium was visible on the WSI, IB/B, PLC, Eo, and Ly were scored as “X”. Otherwise, each finding was given a score of none (0), mild (1), moderate (2), or severe (3) (Figure 2).

Subsequently, a consensus score was obtained for each case. Cases with complete agreement used the evaluated score as the consensus score. Cases with disagreement between the two pathologists used the average of the two scores as the consensus score. For cases where an average could not be taken, i.e., where one of the pathologists scored the slide as X or 0, a third pathologist (JF) reviewed the WSI and a consensus score was reached through discussion with all scoring pathologists.

### 2.3. Ground Truth

Next, ground truth was established by examining pathological reports and clinical courses, and lesions sampled via TBLB were classified into malignant, probably malignant, uncertain, probably benign, and benign (Table 2). The categories were binarized for the purpose of classification model training, with “malignant” and “probably malignant” cases combined into the malignant class and “benign” and “probably benign” combined into the benign class. “Unclear” cases were excluded from the study (Figure 3A).

From the consensus on each pathological finding, odds ratios and AUC values were measured to assess whether they were indicative of malignancy.

### 2.4. Machine Learning Model Development

To create a machine learning model to separate benign vs. malignant states, the data were randomly divided into stratified training and test sets (80%:20%). Given the structure of our data, i.e., many cases lacking data for multiple features (IB/B, PLC, Eo, and Ly), we examined the performance of classifier models which were equipped to handle missing data without imputation. Namely, we chose Decision Tree Classifier, Bagging Classifier, HistGradientBoosting Classifier, and XGBoost Classifier for comparative analysis. We performed 1000 repeats of randomly split training and test sets and took the average performance of each algorithm on these random split repeats. The XGBoost algorithm satisfied our performance requirements, and therefore we proceeded to perform hyperparameter tuning on an XGBoost classifier model with the XGBoost library (v1.7.4) and scikit-learn’s GridSearchCV function using 5-fold cross validation. The hyperparameters tested included all available hyperparameters provided by the XGBoost library, and the values tested were taken from standard accepted ranges. Hyperparameters tested are available for viewing at the linked Github repository. The final model performance values were obtained from randomly split training and test sets, as described above (Figure 3B). Then, 100 repeats of 10-fold cross validation were performed on the training set and the average AUC, F1 score, and accuracy were calculated. Finally, we predicted the malignant vs. benign state of the cases in the held-out test set and calculated the accuracy of these predictions. Findings were ranked by feature importance using the “gain” setting on the XGBoost feature importance function. Gain is defined as the average contribution in the increase of accuracy provided by adding a branch of that feature to the tree, e.g., a gain of 3 means an average gain in classification accuracy of 3% when using that feature as a node on a decision tree. Machine learning analysis was conducted in Python (v3.10.12).

### 2.5. Statistical Analysis

For continuous variables, comparisons between groups were performed with the Kruskal–Wallis rank sum test, and for categorical variables, variables with expected values over five in all categories were evaluated with Pearson’s chi-squared test; otherwise, we used Fisher’s exact test with an averaged *p*-value from 10,000 simulations. Data are either presented as counts and frequencies or means and standard deviations. ROC analysis was performed using the pROC package (v1.18.2). All statistical analysis was performed in R (v4.3.0).

## 3. Results

### 3.1. Clinicopathological Characteristics of Patients

The total number of cases included in this study was 277. The frequency of each ground truth category was 54.5% malignant, 2.5% probably malignant, 9.4% unclear, 12.3% probably benign, and 21.3% benign (Table 2). Malignant and probably malignant were pooled into the malignant group and probably benign and benign were pooled into the benign group. Malignancy was 1.3 times more common in males, while benign cases were approximately 1.5 times more common in females. There were significant differences between malignant and benign cases in age, gender, smoking history, and the number of biopsies taken (Table 1). Higher age, male gender, a history of smoking, and more biopsied specimens correlated with malignant status.

### 3.2. Pathological Findings

IB/B assessed the degree of inflammatory cell infiltration in the respiratory epithelium. For Score 1, the predominant feature was the presence of 3 to 5 lymphocytes per high power field (0.1 mm^2^) beyond the basement membrane, observed in 30 of 135 cases (22.2%). Score 2 exhibited a partial breakdown of the basement membrane, with expanded lymphocytes and neutrophils, primarily around the basal region of the respiratory epithelium, totaling approximately 10 to 20 cells per high power field. In Score 3, there was evident diffuse infiltration of inflammatory cells within the epithelium, and the basement membrane was nearly indistinguishable. Score 0 indicated no inflammatory cell infiltration within the respiratory epithelium.

Ly, PLC, and Eo were evaluated as indicators of the number and density of inflammatory cells present in the stroma within the broncho-vascular bundle. With increasing inflammation, the broncho-vascular stroma thickened, and the spread of inflammation to the surrounding lung parenchyma was observed. OP was characterized by alveoli and alveolar ducts occupied by a fibroblastic plug of the Masson body type. As the score increased, a thickening of the background alveolar septa was observed, accompanied by the presence of mononuclear cell infiltration. FE was observed in the lung parenchyma, not exclusively, but predominantly mixed with some collagenous fibers. With increasing scores, there was a notable aggregation of elastic fibers and a slight increase in dense collagenous fibers, indicative of the presence of chronic alveolar collapse (Figure 2 and Figure 4A). Overall, there were 115 cases which lacked bronchial epithelium of the 251 total cases (45.8%).

### 3.3. Scoring of Pathological Findings

In the Delphi method, the results of the polling showed that FE and Ly were considered to be the most important findings for evaluation according to our polling results, each with a score of 9 (Figure 1). OP and IB/B each had scores of 8 and PLC and Eo had scores of 7. The average score of all polled findings was 5.5.

The odds ratio comparing predictive ability for benign vs. malignant state for each of the six pathological findings indicated that IB/B, PLC, Eo, Ly, and OP showed a tendency toward benign findings, while only FE had a tendency toward malignancy (Figure 5). When analyzing the frequency of lower or higher scores correlating to malignant or benign status, we found a significant correlation between higher FE scores and malignant status, and a significant correlation between higher IB/B, OP, and PLC scores with benign status. Each of these significant findings had a p-value below 0.001, whereas the two that did not, Eo and Ly, had p-values of 0.2 and 0.06, respectively. Although not reaching significance, both findings trended towards higher values being predictive of benign status (Figure 4A).

Among the findings related to benign cases, the AUC values ranged from 0.58 to 0.75. IB/B had the highest AUC for benign findings, at 0.75, and FE had an AUC of 0.66 for malignancy (Figure 4B).

Pairwise correlation analysis using Kendall’s tau coefficient showed that IB/B, PLC, Eo, Ly, and FE were all significantly correlated with each other (Appendix A).

### 3.4. Building a Machine Learning Model to Classify Cases based on Scoring Data

We compared four suitable machine learning algorithms, Decision Tree Classifier, Bagging Classifier, HistGradientBoosting Classifier, and XGBoost. Of these classifiers XGBoost achieved the best performance on accuracy (0.75), ROC AUC (0.753), F1 (0.822), precision (0.744), and recall (0.919), but had the third best performance in execution time (0.059 s) (Appendix A).

We then performed hyperparameter tuning on an XGBoost model which was able to separate cases into malignant and benign with moderate accuracy. The model was trained and cross-validated on a stratified, randomly selected training set of 200 cases. The results of taking the average scores from 100 repeats of 10-fold cross validation on the training set was an AUC of 0.745, F1 score of 0.828, and an accuracy of 0.766. The accuracy of the model on the held-out test set of 51 cases was 0.745, and the AUC was 0.78 (Figure 6A,C). The most important feature was determined to be IB/B, followed by OP, FE, Ly, and Eo, and then PLC (Figure 6B). The analysis of a single randomly selected repeat of 10-fold cross validation showed that the removal of certain features had different effects on performance depending on the fold. The exclusion of IB/B from the classifier resulted in the worst overall regression in performance (Appendix A).

## 4. Discussion

When tumor cells are not detected in small biopsies, these cases are often categorized as non-diagnostic. However, it is noteworthy that there are several pathological findings in these tissues that provide under-recognized clues for predicting malignancy or a benign condition. Utilizing machine learning, we have developed a pathological classification model to distinguish between benign and malignant statuses in TBLB cases without tumor cells. The study results suggest that, for a machine-learning-based classifier, the presence of airway inflammation and organizing pneumonia (OP) suggests a lower likelihood of malignancy and a higher likelihood of a benign status. Conversely, the presence of fibroelastosis (FE) in TBLB specimens suggests a higher likelihood of malignancy. Practically, our data provides pathologists with the opportunity to report these findings in pathology reports as indicative clues for suggesting either a benign or malignant condition, even in inconclusive specimens.

Prior investigations have identified specific pathological changes, including chronic inflammation with fibroplasia, abscess formation, and granulomatous inflammation, as independent predictors of a benign condition [14,15,16,20,21,22]. In addition to these recognized pathological features, the odds ratio and the AUC from this study revealed that IB/B is also associated with a heightened likelihood of a benign condition. This observation aligns with the established fact that bronchitis and bronchiolitis, which are terms encompassing a diverse spectrum of clinical manifestations and syndromes, are mostly exclusively from an inflammatory origin [23,24]. These conditions are predominantly characterized by inflammatory cell infiltration around the airways. Similar to IB/B, our investigation found that Eo, PLC, Ly, and OP exhibit a moderate association with a benign condition.

In contrast to previous studies that predominantly concentrated on a limited set of pathological changes and clinical parameters [14,15,16,20,21,22], our methodology employed a systematic approach, comprehensively reviewing an extensive array of pathological findings to establish their correlation with benign or malignant conditions. These encompassed features such as FE, vascular dilatation, mucus pooling, squamous metaplasia, atypical bronchial epithelium, atypical type II pneumocyte, the presence of necrosis, nuclear debris, atypical adenomatous hyperplasia, and fibrotic background as suggestive features for malignancy, and granuloma, OP, Eo, hemosiderin deposition, Ly, the presence of fungi, amyloid deposition, PLC, IB/B, and the presence of foamy macrophages as suggestive benign features. Our study innovatively employed the Delphi method to analyze this comprehensive list and ultimately distilled it down to six pivotal pathological parameters. This methodology represents a novelty, as it not only systematically scored various features but also selected those with higher frequencies and expected contributions to define benign or malignant conditions.

OP represents a histologic pattern characterized by the aggregation of fibroblasts and collagen, forming a distinctive polypoid structure that involves alveoli and alveolar ducts [25]. This histopathological entity can arise from various causes, such as infections, drug reactions, or connective tissue diseases, or in conjunction with interstitial lung diseases. OP commonly lacks a direct known causative agent, and is thus often categorized as cryptogenic OP [26]. While lung cancer-associated OP has been documented in the literature, its occurrence remains relatively uncommon [27,28]. The etiology of cancer-related OP is not definitively established, with uncertainties persisting regarding whether it represents a secondary manifestation of malignant disease, a consequence of chemo/radiation therapy, or a concomitant presentation. Notably, a study identified OP in approximately 37% of resected lung cancers, particularly associating its presence with squamous cell carcinoma, male gender, and a history of smoking [29]. The mechanism underlying the development of cancer-related OP for squamous cell carcinoma is likely obstructive pneumonia of the peripheral lung. Such can be only seen in the surgically resected lung. Other potential contributing factors include cytokines produced by tumor cells and inflammatory elements within the tumor microenvironment, which may induce pulmonary injury and give rise to OP [28]. Given the lack of contradicting evidence and OP being associated with benign condition in our analysis, its identification in a TBLB sample should be indicative of a non-malignant condition.

FE has frequently been identified in lung adenocarcinoma, with elastosis observed in 80.2% of adenocarcinoma cases [30,31]. The intricate relationship between FE and carcinoma, particularly evident in their early developmental stages, suggests a connection initiated by the detachment of epithelial cells—a phenomenon identified as an initial event in FE formation [32]. In the early phases of cancer initiation, an interplay emerges where inflammatory cells respond to tumor cells, initiating an immune response that leads to the detachment of epithelial cells and subsequently contributes to FE development. While other lung diseases, such as pleuroparenchymal fibroelastosis (PPFE), can also produce FE, their radiological presentation differs. PPFE often manifests radiologically as bilateral pleuroparenchymal thickening with hilar opacities in later stages, and rarely appears as a single nodule as observed in cancer cases [33,34]. Therefore, the presence of radiologic pulmonary nodules in a clinical suspicion of lung cancer, coupled with the identification of FE in transbronchial lung biopsy (TBLB), should heighten the consideration of lung cancer rather than a non-malignant condition.

The XGBoost model performed the best of the four algorithms tested on our dataset other than in execution time, in which it was third fastest. In the XGBoost model, there are primarily three methods for calculating feature importance: gain (the average gain across all splits of the feature), weight (the number of times a feature is used to split the data across all trees), and cover (the average coverage across all splits of the feature). In this study, gain is employed as the measure of feature importance. Figure 6B illustrates the importance of each feature obtained using the XGBoost model, reflecting the degree of influence of separating benign vs. malignant. According to the ranking of feature importance, the most influential variables affecting the prediction results are IB/B, followed by OP, FE, Ly, Eo, and PLC. IB/B is approximately 2.76 times more impactful in increasing model accuracy compared to OP. These data support the concept that inflammatory changes directed towards the airway epithelium are associated with non-malignant airway-related diseases, such as infection. The feature importance analysis results were corroborated by the cross-validation analysis, which showed that performance was most impacted when removing IB/B. By-fold cross validation analysis also showed that performance could differ between folds, which was expected due to the differing distribution of cases with or without bronchial epithelium. For folds which contained large numbers of cases lacking bronchial epithelium, OP and FE were expected to impact overall performance the most.

This study has some limitations. First, for some cases, the follow up period was limited to only 10 months, which is probably insufficient for progression to a malignant disease status. Cases with a more extended follow-up period have the potential to refine the results of the classifier. Furthermore, some of the examined specimens contained malignant cells, which introduced the possibility of bias in the scoring process for these cases.

## 5. Conclusions

Our machine learning classifier provides a robust method for the classification of TBLB specimens, particularly those without identifiable tumor cells. The model achieved a moderately high accuracy and AUC for the dichotomous classification of benign and malignant cases. Odds-ratio analyses on pathological features revealed that IB/B leaned toward a benign condition, while FE was more in favor of malignancy. We believe that these findings will significantly enhance decision making in the evaluation of TBLB samples devoid of tumor cells.

As a future prospect, our goal is to develop artificial intelligence encompassing the entire process from feature extraction to full automation. This initiative aims to enhance patient care by streamlining and automating the entire workflow.

## Figures and Tables

**Figure 1 cancers-16-00731-f001:**
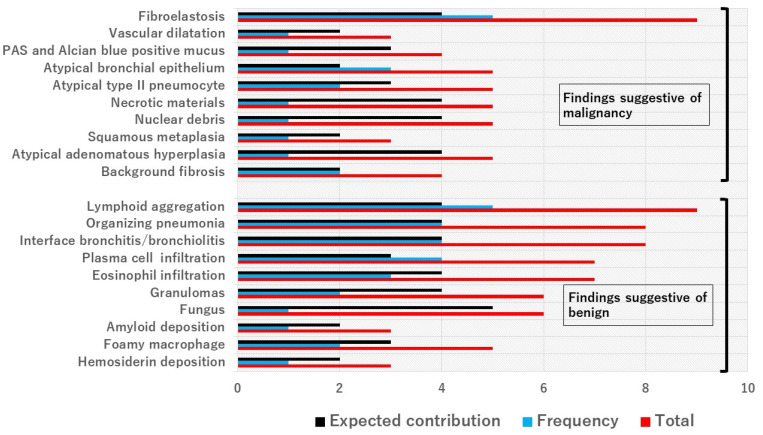
Selection of findings using the Delphi method. Three pathologists (TT, WU, JF) were surveyed to assess what they felt were the most important histological findings suggestive of malignancy or benign status. The pathologists assigned each finding an expected contribution and frequency score. The top six findings were selected (total scores of 7 or above).

**Figure 2 cancers-16-00731-f002:**
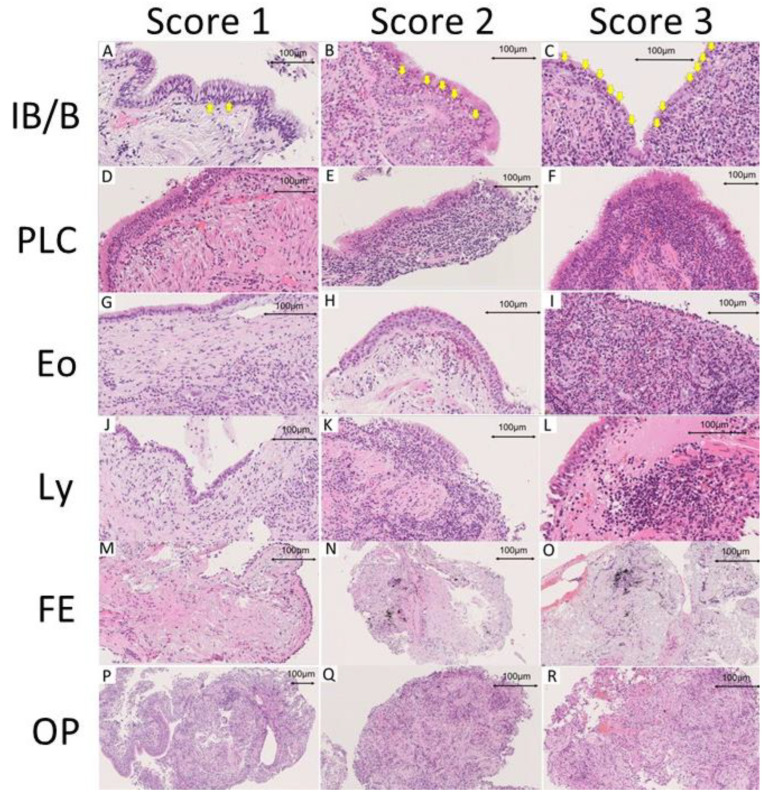
Pathological findings. (**A**–**C**), IB/B; (**D**–**F**), PLC; (**G**–**I**), Eo; (**J**–**L**), Ly; (**M**–**O**), FE; (**P**–**R**), OP. Epithelial infiltration by inflammatory cells (arrow).

**Figure 3 cancers-16-00731-f003:**
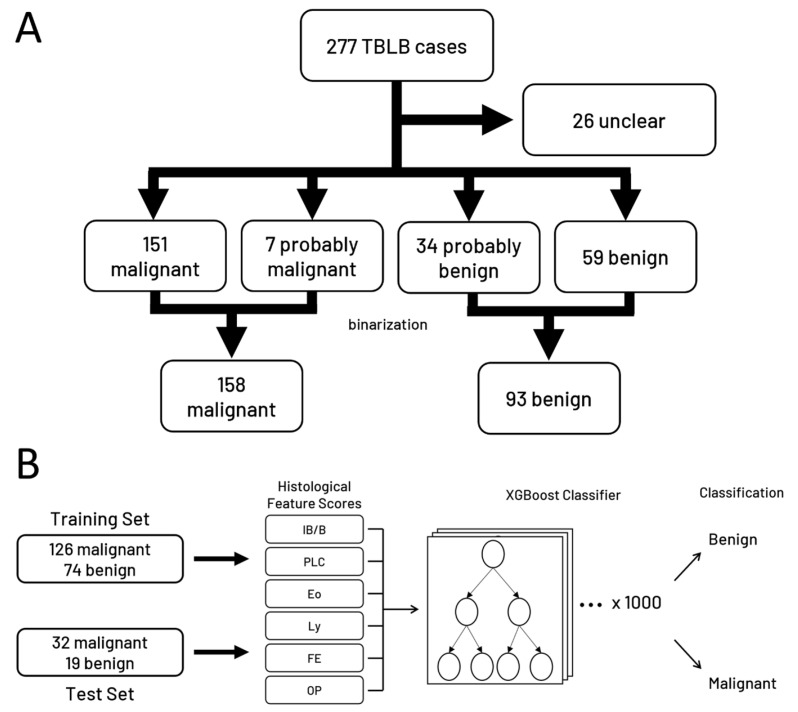
Patient cohort and model structure. (**A**) Flowchart showing case counts separated by disease category. (**B**) Schema describing the structure of the machine learning model used for classification.

**Figure 4 cancers-16-00731-f004:**
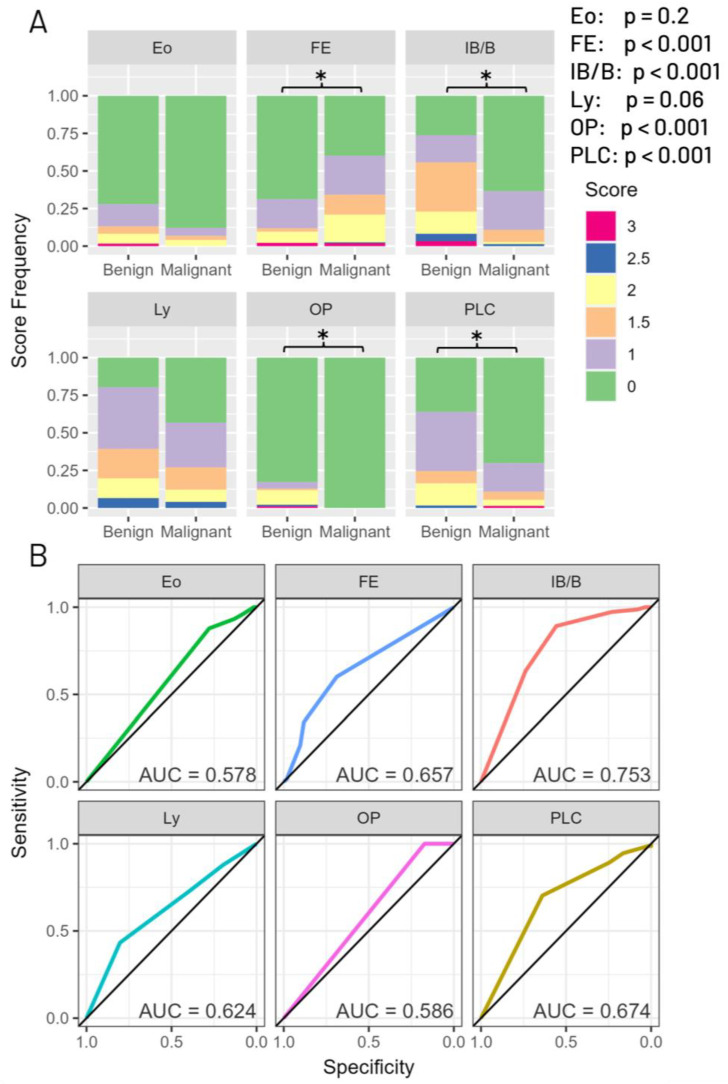
Analysis of each histological finding’s ability to predict malignant vs. benign status. (**A**) Bar plots showing the relative frequencies of each score separated by histological finding. All findings except FE show a correlation between higher scores and benign status, whereas FE shows a significant association between malignancy and the severity of fibroelastosis. Asterisks indicate a significant association between the finding’s scores and disease status (*p* < 0.05). (**B**) ROC curves for scored histological features. Each curve shows the predictive ability for that single feature to predict the malignant or benign status of a case. FE is the only finding which was analyzed as a predictor of malignant status; all other findings were measured as predictors for benign status.

**Figure 5 cancers-16-00731-f005:**
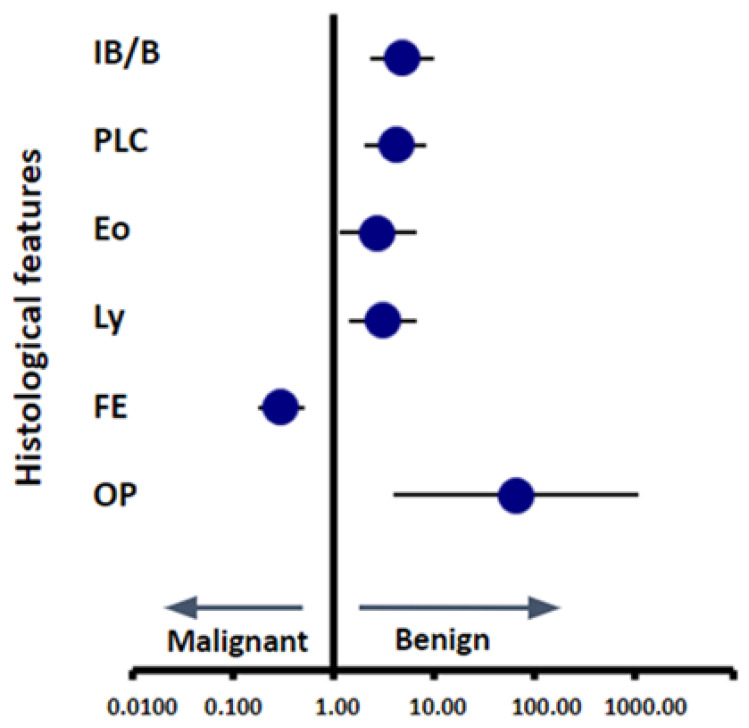
Odds ratio for each histological finding. The circle indicates odds ratio and horizontal lines indicate 95% confidence interval.

**Figure 6 cancers-16-00731-f006:**
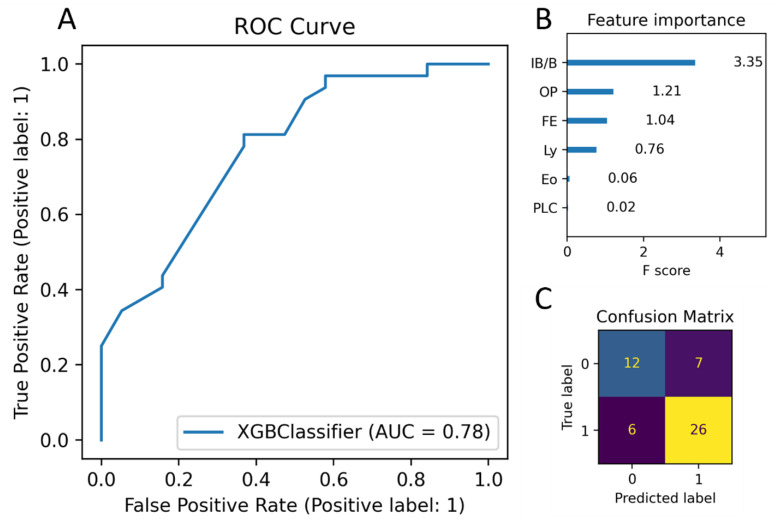
Metrics for decision-tree-based malignant vs. benign classifier based on histological finding scoring data. (**A**) ROC curve for the XGBoost classifier model. AUC = 0.78. (**B**) Feature importance chart for trained model. Feature importance calculated as “gain”, i.e., the average percentage that accuracy increases by when using each feature in the decision tree. (**C**) Confusion matrix for the held-out test set of 51 cases; 1 = malignant, 0 = benign. Dark purple means values comprised between 0-10, dark blue is between 11 and 20, and yellow beyond 21.

**Table 1 cancers-16-00731-t001:** Demographic and biopsy data. Continuous variables compared with the Kruskal–Wallis test and categorical variables were evaluated with either Pearson’s chi-squared test or Fisher’s exact test.

Variable	Malignant (*n* = 158)	Benign (*n* = 93)	*p*-Value
Age	73.7 ± 8.0 (74)	68.5 ± 13.0 (71)	<0.01
Gender M/F	93/65	41/52	0.02
Smoking Index	746 ± 660 (662)	341 ± 528 (0)	<0.01
Number of specimens	7.8 ± 3.3 (7)	5.7 ± 2.6 (5)	<0.01
Location (lobe)			>0.05
Right upper	54 [21%]	29 [11%]	
Right middle	10 [4%]	14 [6%]	
Right lower	38 [15%]	17 [7%]	
Left upper	36 [14%]	19 [8%]	
Left lower	20 [8%]	14 [6%]	

Mean ± SD (median) [%]; 26 out of 277 cases were excluded as they were considered to be unclear for malignant vs. benign after the review of clinical and follow-up data.

**Table 2 cancers-16-00731-t002:** Definitions of ground truth categories.

Judgement	Definition	N
Malignant	Pathologically confirmed malignancy at the time of biopsy, follow-up biopsy, cytology, or in subsequent surgical materials	151
Probably Malignant	Clinically diagnosed and treated as malignant without pathological evidence of malignancy	7
Unclear	No pathological evidence of malignancy, with a clinical diagnosis of difficulty in determining malignancy and ongoing follow-up	26
Probably Benign	No pathological evidence of malignancy, with a clinical diagnosis of benign disease and ongoing follow-up	34
Benign	No pathological evidence of malignancy, clinically diagnosed as benign, and treated (e.g., antibiotics for infection) or discharged	59

## Data Availability

The source code for the machine learning classifier can be found at https://github.com/eokoshi/AI-TBLB-Assist (accessed on 7 February 2024).

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
