# Peer review of "Machine-Learning-Based Classification Model to Address Diagnostic Challenges in Transbronchial Lung Biopsy"

_cancers, 2024, doi:10.3390/cancers16040731_

Round 1

Reviewer 1 Report

Comments and Suggestions for Authors

The article entitled “Machine Learning-Based Classification Model to Address Diagnostic Challenges in Transbronchial Lung Biopsy” is well-written and, from my point of view, would be of interest of readers of Cancers. In spite of this, there are certain issues that should be improved before its publication. Those issues are as follows:

In the final part of the introduction, I suggest to include a paragraph that describes the layout of the article.

In Table 1 in the variable location (lobe) the percentage that each category represents should be included.

Figure 1. The figure is fuzzy. I do not really understand how to distinguish blue from red and interprete results. Please, change it. I suggest using one bar for red and another for blue.

Table 2: what are those numbers? It should be indicated in the header of the referred table.

In section 2.5. Statistical Analysis the XGBoost model is included. Please consider it treating apart as it is a machine learning methodology and a more in-depth explanation is required.

Figure 4 is fuzzy. Please, insert it again in a higher quality.

Author Response

Comments 1: In the final part of the introduction, I suggest to include a paragraph that describes the layout of the article.

Response 1: Thank you for pointing this out. We included a paragraph that describes the layout of the article as below (Page 2, Line 43-47).

In this manuscript, we initially extract cases of TBLB for pulmonary nodular lesions. We then establish histological candidates to be scored using the Delphi method. Following this, we obtain scores from multiple pathologists, develop a machine learning algorithm using these scores, and evaluate the algorithm's effectiveness in classifying TBLB samples as either benign or malignant.

Comments 2: In Table 1 in the variable location (lobe) the percentage that each category represents should be included.

Response 2: In Table 1 in the variable location (lobe), we included the percentage.

Comments 3: Figure 1. The figure is fuzzy. I do not really understand how to distinguish blue from red and interprete results. Please, change it. I suggest using one bar for red and another for blue.

Response 3: We modified the Figure 1. Figure 1 illustrates the Delphi method, utilizing three bars for each observation: one for expected contribution (black), another for frequency (blue), and the third representing their combined total (red).

Comments 4: Table 2: what are those numbers? It should be indicated in the header of the referred table.

Response 4: Those numbers are “N”. We added N in the header of the referred table.

Comments 5: In section 2.5. Statistical Analysis the XGBoost model is included. Please consider it treating apart as it is a machine learning methodology and a more in-depth explanation is required.

Response 5: Description of feature importance was moved to the “Machine Learning Model Development” section of the Materials and Methods (Page 7, Paragraph 2.4). We also provided additional context and explanation as requested. 

Comments 6: Figure 4 is fuzzy. Please, insert it again in a higher quality.

Response 6: We inserted Figure 4 in a higher quality.

4. Response to Comments on the Quality of English Language

Point 1: English language fine. No issues detected

Response 1: Thank you. Our double first author, EO, is a native English speaker.

Reviewer 2 Report

Comments and Suggestions for Authors

The paper is pivotal to the aims of the Journal and presents a very important topic of interest for data scientists and clinicians at large, potentially improving life expectancy and quality of a vast amount of patients even worldwide.

However, there are a few points to be acknowledged before going further with the acceptance of the manuscript, as mentioned below:

- The Introduction should go more into depth about the existing literature about the usage of AI/ML in the clinical scenario of the paper

- Ethical approval should be acknowledged also in the main text

- As for ML model selection, why being just limited to the XGBoost model instead of trying other ML models? And, also other models can manage missing data that can be also managed with dedicated algorithms. Please, discuss.

- Please, discuss (also in the text when presenting Table 1) the discrepancies between numbers in term of recruited individuals and those actually included in the dataset (e.g., 277 against 251).

- In the Discussion section, I suggest avoiding personal considerations unless duly supported by literature citations.

- Overall, English language and grammar should be improved throughout the manuscript.

Comments on the Quality of English Language

English language and grammar should be improved throughout the manuscript.

Author Response

Comments 1: The Introduction should go more into depth about the existing literature about the usage of AI/ML in the clinical scenario of the paper

Response 1: Thank you for pointing this out. We added the usage of AI/ML in the clinical scenario in the introduction (Page 2, line 32-37).

Comments 2: Ethical approval should be acknowledged also in the main text

Response 2: Agree. We added Ethical approval. The approval number in the IRB is M2021-315 (Page 3, Paragraph 2.1, Line 5-7).

Comments 3:  As for ML model selection, why being just limited to the XGBoost model instead of trying other ML models? And, also other models can manage missing data that can be also managed with dedicated algorithms. Please, discuss.

Response 3: Due to the design of our study, many of our samples had missing data. Cases lacking bronchial epithelium on histological imaging could not be scored for features other than FE and OP. Therefore, it was not appropriate for our dataset to handle missing data via imputation or similar methods. The only algorithms which were able to handle this type of data native to the scikit-learn library (v1.4.0) were the Decision Tree, HistGradBoosting, and Bagging algorithms. After further research, we found that XGBoost could also handle this type of data due to its ternary tree structure. XGBoost was found to have the best performance out of the 4 algorithms tested. Based on the reveiwer’s suggestion, we modified the text (Page 7, Paragraph  2.4)

Comments 4: Please, discuss (also in the text when presenting Table 1) the discrepancies between numbers in term of recruited individuals and those actually included in the dataset (e.g., 277 against 251).

Response 4: In the Patient cohort, although initially stated as 277 consecutive TBLB cases, Table 1 provides an explanation for a total of 251 cases. The discrepancy of 26 cases is clarified in Figures 2 and 3A. These 26 cases were categorized as 'Unclear' in the Ground Truth and were not included in the scope of this study. Therefore, the total number was 277 in the description. Explanation was added for Table 1 to avoid the confusion.

Comments 5: In the Discussion section, I suggest avoiding personal considerations unless duly supported by literature citations.

Response 5: Agree. We have transitioned the expression from personal reflections to objective analysis in the Discussion (Page 11, Line 2, 6, 10), (Page 12, Line 5-6, 25,), (Page 13, Line 14).

Comments 6: Overall, English language and grammar should be improved throughout the manuscript.

Response 6: Our double first author, EO, is a native English speaker, and we have carefully revised the manuscript to enhance the overall English language and grammar (Page 8, Paragraph 3.2, Line 2), (Page 13, Paragraph 4, Line 23).

4. Response to Comments on the Quality of English Language

Point 1: Moderate editing of English language required

Response 1: We addressed this in Response 6.

Reviewer 3 Report

Comments and Suggestions for Authors

There are several modifications required in the manuscript.My comments are as follows :

1. Could the authors please provide a detailed account of their specific contributions in comparison to the published literature? It would be beneficial if the distinctiveness of their contributions could be justified.

2.  It is always recommended to apply 10-fold CV for unbiased and reliable prediction results, which is not possible due to the random split of data during training and testing. It is kindly requested to consider these recommendations and add results and discussion in the revised version with reference to the suggested references provided.

  1. 10.1088/1402-4896/acae49
  2. https://doi.org/10.1080/27684830.2023.2201015

3. Did cross-validation help identify whether the impact of redundant features was consistent across different subsets of the dataset?  Were there instances where the inclusion or exclusion of redundant features affected model performance differently in various cross-validation folds?

4. Comprehensive details related to hyperparameter selection of ML models need to be included. What are the criteria for selecting the hyperparameters? Please refer to recently published papers and add a discussion in the revised version.

  1. https://doi.org/10.3390/pr11020349
  2. https://doi.org/10.1145/3610536 

5. Inclusion of a confusion matrix and essential metrics such as Accuracy, Precision, Recall, and F-score for all the machine learning models examined in this study is imperative. Additionally, it is essential to incorporate a comparative analysis in the Results and Discussion section, which is missing in the submitted manuscript.

6. Figures quality need to be improved. Please add high resolution figures.

7. Conclusion section is very short. Kindly refer suggested references and add necessary details and also add future scope.

Comments on the Quality of English Language

English Language is ok.

Author Response

Comments 1: Could the authors please provide a detailed account of their specific contributions in comparison to the published literature? It would be beneficial if the distinctiveness of their contributions could be justified.

Response 1: Thank you for pointing this out. We added in author contributions including machine learning part.

Comments 2: It is always recommended to apply 10-fold CV for unbiased and reliable prediction results, which is not possible due to the random split of data during training and testing. It is kindly requested to consider these recommendations and add results and discussion in the revised version with reference to the suggested references provided.

  1. 10.1088/1402-4896/acae49
  2. https://doi.org/10.1080/27684830.2023.2201015

Response 2: We thank the reviewer for their suggestion. 10-fold CV was performed after random train-test splitting, and performance metrics were reported in section 3.4 (Results/Building a Machine Learning Model to Classify Cases based on Scoring Data). 100 repeats of 10-fold CV were conducted without re-splitting the train-test data. In other words, the 100 repeats of CV were performed on the same training set. This was done to attempt to reduce the bias and replicability of these results.  

Comments 3: Did cross-validation help identify whether the impact of redundant features was consistent across different subsets of the dataset?  Were there instances where the inclusion or exclusion of redundant features affected model performance differently in various cross-validation folds?

Response 3: Thank you to the reviewer for bringing up this interesting inquiry. According to the results of pairwise Kendall rank coefficient testing, IB/B, PLC, Eo, Ly, and FE were significantly correlated with each other. Upon conducting 10-fold CV with each of these findings removed, We found that removing IB/B worsened model performance the most, whereas removing Eo and PLC affected performance the least, which aligns with our feature importance analysis. Removing each of the other findings somewhat worsened performance, but not as drastically as IB/B. We found no clear differences in performance between CV folds which correlated specifically to one feature, other than with IB/B which saw a significant decrease in performance when removed relative to the other tests in folds 1, 2, 4, 6, and 8. Correlation and CV analysis was included as supplemental data. 

Comments 4: Comprehensive details related to hyperparameter selection of ML models need to be included. What are the criteria for selecting the hyperparameters? Please refer to recently published papers and add a discussion in the revised version.

  1. https://doi.org/10.3390/pr11020349
  2. https://doi.org/10.1145/3610536 

Response 4: Hyperparameter tuning was conducted on all available hyperparameters provided by the XGBoost python library. Hyperparameter tuning was conducted using sklearn’s GridSearchCV function, which means that each hyperparameter combination was exhaustively tested and the combination with the greatest performance was selected. The appropriate section of the Materials and Methods was furnished with further information(Page7, Paragraph 2.4). 

Comments 5: Inclusion of a confusion matrix and essential metrics such as Accuracy, Precision, Recall, and F-score for all the machine learning models examined in this study is imperative. Additionally, it is essential to incorporate a comparative analysis in the Results and Discussion section, which is missing in the submitted manuscript.

Response 5: We thank the reviewer for their input. We have included the recommended data as supplementary tables and figures, as well as including more discussion on this topic in the results and discussion sections (Page 10, Paragraph 3.4), (Page 13, Paragraph 4, Line 4-22). 

Comments 6: Figures quality need to be improved. Please add high resolution figures.

Response 6: We inserted Figures in a high quality.

Comments 7: Conclusion section is very short. Kindly refer suggested references and add necessary details and also add future scope.

Response 7: In conclusion, we have incorporated additional sentences addressing the future scope of our work. (Page 13, Paragraph 5, Line 8-10).

4. Response to Comments on the Quality of English Language

Point 1: Minor editing of English language required

Response 1: Our double first author, EO, is a native English speaker.

Reviewer 4 Report

Comments and Suggestions for Authors

Why authors only use the XGBoost? When there are other state of the art models available. 

Why authors focused on the Transbronchial Lung Biopsy?

Below papers have some interesting implications that you could discuss in your introduction and how it relates to your work.

GV Eswara Rao, B Rajitha, Parvathaneni Naga Srinivasu, Muhammad Fazal Ijaz, Marcin Woźniak. Hybrid framework for respiratory lung diseases detection based on classical CNN and quantum classifiers from chest X-rays. Biomedical Signal Processing and Control

A Deep Transfer Learning Approach For Covid-19 Detection And Exploring A Sense Of Belonging With Diabetes. Ijaz Ahmad, Arcangelo Merla, Babar Shah, Ahmad Ali Alzubi, Mallak Ahmad AlZubi, Farman Ali. Frontiers in Public Health

It would be interesting if the authors report the trade-off compared to other methods especially the computational complexity of the models. Some techniques require more memory space and take longer time, please elaborate on that. 

What are the practical implications of your research? 

I would suggest author to use another dataset. Another dataset would consolidate the work if the authors obtain a consisting results. please elaborate on that. 

Mention the limitations of the present work?

Comments on the Quality of English Language

..

Author Response

Comments 1: Why authors only use the XGBoost? When there are other state of the art models available. 

Response 1: Thank you for pointing this out. We tested four algorithms and found that XGBoost performed best on our dataset. We have included more information regarding this testing in the Materials and Methods, Results, and Discussion sections. Thank you for your question. 

Comments 2: Why authors focused on the Transbronchial Lung Biopsy?

Below papers have some interesting implications that you could discuss in your introduction and how it relates to your work.

 • GV Eswara Rao, B Rajitha, Parvathaneni Naga Srinivasu, Muhammad Fazal Ijaz, Marcin Woźniak. Hybrid framework for respiratory lung diseases detection based on classical CNN and quantum classifiers from chest X-rays. Biomedical Signal Processing and Control

• A Deep Transfer Learning Approach For Covid-19 Detection And Exploring A Sense Of Belonging With Diabetes. Ijaz Ahmad, Arcangelo Merla, Babar Shah, Ahmad Ali Alzubi, Mallak Ahmad AlZubi, Farman Ali. Frontiers in Public Health

Response 2:  Among the several biopsy approaches for lung nodules, transbronchial lung biopsy (TBLB) is the most common. And many TBLB specimens present challenges in distinguishing between benign and malignant conditions. The motivation for our research stemmed from exploring the possibility of using AI/ML to classify these challenging specimens. We have incorporated the referenced paper you provided in our references (Page 2, Line 37).

Comments 3: It would be interesting if the authors report the trade-off compared to other methods especially the computational complexity of the models. Some techniques require more memory space and take longer time, please elaborate on that.

Response 3: Thank you for your suggestion. We have included analysis of the comparative execution time for each of the four evaluated models (Supplemental Table 1).

Comments 4: What are the practical implications of your research? 

Response 4: We think our machine learning classifier effectively categorizes TBLB specimens, especially those lacking identifiable tumor cells. These results are expected to greatly improve decision-making in assessing TBLB samples without tumor cells. That is the practical implication of our research. This was highlighted in the text (Page 13, Paragraph 5, Line 1-7).

Comments 5: I would suggest author to use another dataset. Another dataset would consolidate the work if the authors obtain a consisting results. please elaborate on that. 

Response 5: Thank you for pointing this out. Due to the limitation under the current IRB approval, we would like to continue the research in the future project using an objective dataset, excluding the limitations outlined in the paper.

Comments 6: Mention the limitations of the present work?

Response 6: Agree. We mentioned limitations (Page 13, Paragraph 4, Line 23-28).

4. Response to Comments on the Quality of English Language

Point 1: Minor editing of English language required

Response 1: Our double first author, EO, is a native English speaker.

Round 2

Reviewer 3 Report

Comments and Suggestions for Authors

Authors have revised manuscript based on suggested comments.

Comments on the Quality of English Language

English Language is ok.

Reviewer 4 Report

Comments and Suggestions for Authors

.